# Development of General Exposure Factors for Risk Assessment in Korean Children

**DOI:** 10.3390/ijerph17061988

**Published:** 2020-03-18

**Authors:** Hyojung Yoon, Sun-Kyoung Yoo, Jungkwan Seo, Taksoo Kim, Pyeongsoon Kim, Pil-Je Kim, Jinhyeon Park, Jung Heo, Wonho Yang

**Affiliations:** 1Environmental Health Research Division, National Institute of Environmental Research, Incheon 22733, Korea; hyojay97@korea.kr (H.Y.); ysk1004@korea.kr (S.-K.Y.); jkseo2001@korea.kr (J.S.); taksoo@korea.kr (T.K.); iprecious@korea.kr (P.K.); newchem@korea.kr (P.-J.K.); 2School of Environmental Engineering, University of Seoul, Seoul 02546, Korea; 3Department of Occupational Health, Daegu Catholic University, Gyeongbuk 42472, Korea; venza11@naver.com (J.P.); u2zero@naver.com (J.H.)

**Keywords:** children, exposure factors, environmental hazards, risk assessment

## Abstract

There has been an increasing need for the risk assessment of external environmental hazards in children because they are more sensitive to hazardous chemical exposure than adults. Therefore, the development of general exposure factors is required for appropriate risk assessment in Korean children. This study aimed to determine the general exposure factors among Korean children aged ≤18 years. We developed the recommended exposure factors across five categories: physiological variables, inhalation rates, food and drinking water intake, time–activity patterns, and soil and dust ingestion. National databases were used, and direct measurements and questionnaire surveys of representative samples were performed to calculate the inhalation rate, water intake, and soil ingestion rate. With regard to the time–activity patterns, the daily inhalation rates ranged from 9.49 m^3^/day for children aged 0–2 years to 14.98 m^3^/day for those aged 16–18 years. This study found that Korean children spent an average of 22.64 h indoors, 0.63 h outdoors, and 0.73 h in-transit on weekdays. The general exposure factors of Korean children were studied for the first time, and these results could be used to assess children’s exposure and risk. They also suggest the differences compared with the results of international results.

## 1. Introduction

Assessment of exposure to environmental pollutants requires a wide range of exposure factors to evaluate air, food, and water intakes and frequency and duration of exposure, together with their concentrations in air, water, food, and soil [1]. Since individual exposure to hazardous substances may significantly vary depending on the exposure factors related to human behaviors and characteristics, a number of aspects should be considered for accurate exposure estimation.

Children’s pattern of exposure to environmental hazards significantly differs from that of adults. Children can be exposed to hazardous substances in the following ways: breastfeeding, hand-to-mouth contact, and their dynamic developmental physiology [2]. Children are more likely to be exposed to environmental pollutants because their inhalation rates and food/water intakes per body weight are higher than those of adults. According to a World Health Organization (WHO) report, children are exposed to 2.3 times more air, 4.8 times more drinking water, and 6.1 times more foods per body weight than adults [3]. As children are particularly sensitive to chemical exposure, careful attention should be paid to the development and assessment of exposure factors in children. Young children are also known to have higher exposure to pesticides as they frequently come in contact with soil and dust on surfaces, toys, and other objects [4]. Previous studies have found that exposure to hazardous substances at young or early juvenile age disrupts neural development or causes neurobehavioral damages [5,6].

The United States Environmental Protection Agency (EPA) has created a handbook on exposure factors for children by compiling related data from various sources [7]. China has developed exposure factor resources for children, but these data are currently not available in English. While Korea published an exposure factor handbook for adults in 2014 [8], there has been an increasing need to separately assess various factors related to exposure assessment in sensitive groups such as children. Recently, in Korea, children’s risk of exposure to environmental pollutants was potentially over/underestimated in risk assessment by applying default values or exposure factors used in other countries, including the United States. Additionally, exposure factors that represent Korean children’s unique characteristics should also be determined to ensure accurate risk assessment and to better establish relevant environmental national standards [9]. We have been studying the general exposure factors in Korean children since 2013; the results were published in 2017 [10]. In contrast, this study was carried out for international comparative evaluation by adding updated national statistics and new experiments, such as those for the rates of soil and dust ingestion and unintentional exposure through hand-to-mouth contact.

This study aimed to propose the representative or recommended values for exposure factors by analyzing the characteristics and behavioral patterns of Korean children and to provide a foundation for more accurate exposure and risk assessment. This would be helpful to establish relevant environmental regulations, standards, and policies for the first time in Korea.

## 2. Materials and Methods

In this study, we proposed the means, standard deviations, percentages, and other statistical values for Korean children aged ≤18 years for the following five categories: physiological variables, inhalation rates, food and water intake, time–activity patterns, and soil and dust ingestion. The latest data from national organizations, such as Statistics Korea, were used to develop the general exposure factors, and measurements and questionnaire surveys were performed if there were no existing data.

### 2.1. Categories of Exposure Factors and Age Groups

We obtained the representative or recommended values of the following five categories: physiological variables (body weight and surface area), inhalation rate, food (grains, meat, fish and shellfish, vegetables, fruits, and processed foods) and water intake, time–activity patterns (times spent at microenvironments and in vehicles), and non-dietary ingestion (soil and dust ingestion).

Each legislation has different age criteria for children: ≤13 years in the Environmental Health Act of the Ministry of Environment (MOE) [11] and <18 years in the Child Welfare Act of the Ministry of Health and Welfare in Korea [12]. Children are internationally defined as individuals aged <18 years. Therefore, we categorized children into several similar time–activity groups: infant group (age: <1 year), toddler group (age: 1–2 years), preschooler group (age: 3–6 years), elementary school student group (age: 7–12 years), and middle/high school student group (age: 13 and 18 years). Each group was divided into subgroups as necessary to calculate the recommended values for each exposure factor.

### 2.2. Physiological Exposure Factors

The average daily doses are typically determined by calculating the level of exposure to environmental hazards through breathing and dermal contact based on the specific group’s average body weight. Children, in particular, need to use the recommended values because their weight varies considerably with age.

We used the body weight data of 6290 children aged >1 year from the Korea National Health and Nutrition Examination Survey (KNHANES, 2013–2016) and body weight data of 406 infants aged <12 months from the 5th Size Korea survey conducted by the Korean Agency for Technology and Standards under the Ministry of Industry and Energy (2003–2004).

Body surface area (BSA) is one of the major exposure factors required for estimating the extent of dermal exposure to chemicals. As BSA measurement requires considerable time and efforts, regression equations have been proposed, in which the body weight and height of a specific population size were directly measured to predict BSA [13]. This study used a regression equation formulated based on actual measurements of 305 children (aged 8–18 years) reported by the Korea Ministry of Food and Drug Administration [14]. The recommended BSA values were calculated for different age groups using their average weights (W) and heights (H):BSA (cm^2^) = 0.007331 × W0.425 × H0.725

### 2.3. Inhalation Rate

During the study, there were no data available to calculate the recommended levels for inhalation rate. Thus, the present study performed a direct measurement and data analysis. We measured the body weights and respiratory volumes of 262 children aged ≥5 years who were grouped into preschoolers, elementary school students, and middle/high school students, considering sex and age in 2017. With regard to respiratory volumes, we measured children’s oxygen consumption, metabolism, and ventilation at different activity phases, including resting (in a sedentary state), walking (at 2–3 km/h), and slow running (at 3–4 km/h) for 5–10 min, respectively, using a gas analyzer (Quark b2, COSMED, Italy). Moreover, physical activity patterns were recorded every 10 min using a time–activity diary for 3 days. The respiratory volumes according to the different activity phases were combined with the physical activity patterns. Subsequently, daily inhalation rates (m^3^/day) were calculated [15]. For the group aged <4 years, their daily inhalation rates were estimated using a regression equation based on the correlation between inhalation rate and body weight because measurements of inhalation rates were unattainable.

### 2.4. Food and Drinking Water Intake

The KNHANES (http://www.mohw.go.kr/) is one of the key surveys that represent Koreans’ food and nutrient intake. This study analyzed the data of 5,823 children or adolescents aged ≤18 years among the participants of the 2013–2016 surveys. We compiled a list of food items consumed by the surveyed group and categorized them into 14 food groups, including grains, vegetables, fruits, meat, fish/shellfish, and seaweed. To adjust for water loss from different cooking methods, we applied different criteria for calculating consumption depending on whether each food group was consumed as cooked or raw ingredients. This study recategorized the food groups according to the criteria used in the previous study [7].

### 2.5. Time–Activity Patterns

To analyze children’s time–activity patterns, a questionnaire survey using a time–activity diary was conducted on 2080 children aged ≤9 years according to sex in 2013–2014. The children’s parents recorded the information using a time–activity diary, and activity information was collected in 10-min intervals. The activities were organized according to the following classification: main activity, additional activity, transportation, indoors, and outdoors. Children aged less than 10 years were selected using a stratified sampling method with the proportional and square root allocation for the country. Their time–activities were analyzed by sex, age, residential area, time (weekdays/weekends), and season. The time–activity pattern data of 27,716 children aged 10–18 years were obtained using the time-use survey method of the Korea National Statistical Office (http://www.survey.go.kr/lifestyle). The diary classification of this survey included residential indoor, school classroom, and transportation by sex and age (weekdays/weekends).

### 2.6. Hand-To-Mouth and Object-To-Mouth Activities

A questionnaire survey and videotaping of children’s mouthing behaviors, such as hand-to-mouth and object-to-mouth behaviors were conducted at home and in a kindergarten school where they spent the majority of their time during 2014–2015. We surveyed three metropolitan cities (Incheon, Gwangju, and Daegu), and samples were selected based on the 2014 Statistics of Registered Population of the Ministry of the Interior and Safety. The surveyed children were divided into two groups: children aged 1–2 years and those aged 3–6 years. Two to four camcorders were set up at each daycare center to investigate the frequency (contacts/h) and time (sec/contact) of 147 children’s hand-to-mouth and object-to-mouth activities per hour [16,17]. In their houses, the children’s behaviors were observed with a video recording on a mobile phone for 1 h. The children’s contacts with pacifiers were excluded. We analyzed the video recordings of children’s mouthing behavior patterns and crosschecked the findings to reduce errors between analyzers.

### 2.7. Soil and Dust Ingestion

Children may be exposed to polluted soil or dust when they unintentionally put their hands or objects into their mouths. The present study investigated the soil intake of children and calculated its recommended levels based on the results. The limiting tracer method was used to estimate children’s soil and dust ingestion rates [18]. First, the concentrations of tracers in children’s feces, neighborhood soil, and indoor dust at home were measured. For the tracers used in the measurement, aluminum, silicon, titanium, and zirconium were selected because of their low levels of consumption via food and their low intestinal absorption rates, although they are abundant in soil. As a result, soil intakes were calculated based on the aluminum concentration. Based on the assumption that soil ingestion could not exceed the lowest concentration value among the tracers, children’s soil and dust ingestion was estimated according to the equation by Binder et al. (1986) [19]:(1)$$\mathrm{Ti},\;e=\frac{\;\mathrm{fi},e\;\times \;\mathrm{Fi}}{\mathrm{Si},e}$$
where Ti, e is subject i’s estimated ingestion of soil and dust based on the concentration of tracer e (g/day), f i, e is the concentration of tracer e in subject i’s feces (mg/g), F i is the weight of subject i’s dried feces (g/day), and S i, e is the concentration of tracer e in dust and soil in subject i’s living environment (mg/g).

A total of 113 children aged 1–6 years living in metropolitan cities (Incheon, Daegu, and Gwangju) and small/medium-sized cities (Cheongju, Gyeongsan, and Yangpyeong) participated in the investigation. Their concentrations of tracers in children’s feces, neighborhood soil, and indoor dust were analyzed in 2016–2017. The feces of children for 48 h and the soil near their major activity places (home and day-care center) were collected to analyze the concentration of tracers. The collected soil was treated using the EPA’s process testing method, and indoor dust collected using polyvinyl chloride filters was digested using a microwave [7]. The treated samples were analyzed using inductively coupled plasma atomic emission spectroscopy (Agilent Tech.). To adjust for dietary and other sources of tracer intakes, three children who did not engage in outdoor activities formed the control group. Ultimately, the recommended values for soil and dust ingestion were calculated by accounting for the background concentration of each tracer.

### 2.8. Statistical Analysis

All data were statistically analyzed using the t-test and the one-way analysis of variance. Regression analysis was used to analyze the association between respiratory volumes and body weights. All data were analyzed using the Statistical Package for the Social Sciences version 18.0 (IBM, USA). This study was approved by the Institutional Review Board (CUIRB-2016-0104) at Daegu Catholic University.

## 3. Results

### 3.1. Physiological Exposure Factors

We grouped the children aged <12 months into four age groups along with six other age groups and proposed the average body weights and percentages for the ten age groups.

There were significant differences in body weight between boys and girls (p < 0.05). The average body weights of children aged ≤3 months, 3–6 months, <9 months, and <12 months were 5.6 kg, 7.5 kg, 8.7 kg, and 9.7 kg, respectively. The average body weights of children aged 1–2 years, 3–6 years, 7–9 years, 10–12 years, 13–15 years, and 16–18 years were 12.2 kg, 18.8 kg, 30.0 kg, 43.8 kg, 57.4 kg, and 62.4 kg, respectively (Table 1).

The recommended values and statistics for body surface area by age group are shown in Table 2. The average BSA of infants less than 3 months was 60% compared with that of infants aged 1 year (Figure 1). BSAs increased rapidly until the age of 12 months. However, it gradually increased after.

### 3.2. Inhalation Rate

The inhalation rates of children aged ≥5 years, which may be a feasible age for survey, were measured. The findings indicated a significantly positive correlation of hourly inhalation rates with ages and activity levels across both sexes. Additionally, boys exhibited significantly higher inhalation rates than girls across all ages, as shown in Table 3. When estimating the short-term inhalation rates of children aged <4 years, a regression equation was formulated by analyzing the association between inhalation rates and body weights of children aged ≥5 years (Figure 2). The daily inhalation rates were estimated based on time–activity patterns and short-term inhalation rates. Table 4 lists the average values of the analyzed groups aged 0–2, 3–6, 7–9, 10–12, 13–15, and 16–18 years. The daily inhalation rate of each age group ranged between 9.23 and 16.15 m^3^/day (0.24–0.90 m^3^/kg/day).

### 3.3. Food and Drinking Water Intake

The intakes and percentage of different types of food across age groups are shown in Table 5. The combined data of male and female children were reported. The food with the highest intake was grains, followed by vegetables, dairy products, fruits, meat, and fish and shellfish. Average water intake was 800 mL per day. Girls consumed more fruits than boys, while boys consumed higher amounts of grains, vegetables, meat, dairy products, and mixed/processed food. Food and water intakes increased with age. However, the 13–to-18-year age group reported the lowest intake of fruits and dairy products. The average and 95th percentile consumption rates of children and adolescents for consumers only are shown in Table 6. The average intake of fabricated foods for consumers only (115.36 g/day) was seven times larger than that for all children (15.96 g/day).

The survey of infants and toddlers aged ≤3 years from 2014 to 2016 (n = 808) showed that 90.4% of the surveyed children consumed breast milk, 26.8% consumed only breast milk, and 73.2% consumed formula milk. The average length of time for breastfeeding was 9.6 months. Meanwhile, weaning from breastfeeding, and eating foods and drinking regular milk started at 6.2 months and 14.5 months, respectively.

### 3.4. Time–Activity Patterns

The time–activity patterns of children by sex, age, and weekdays/weekends were analyzed based on the results of a survey conducted in children aged 0–9 years and times-use surveys conducted by Statistics Korea in children aged 10–18 years. The recommended values by age and weekday and weekend are shown in Table 7. For older children, the time spent at home on weekdays and weekends decreased, while the time spent in other indoor environments such as schools and other educational facilities increased. On weekends, low-grade elementary school students (aged ≤9 years) spent more times performing outdoor activities than children in the other groups.

### 3.5. Hand-To-Mouth and Object-To-Mouth Activities

Young children are more likely to be exposed to hazardous substances than adults as they tend to put objects (toys, books, or stationery) in their mouths and hands. In this study, we investigated the mouthing behaviors of toddlers (aged 1–2 years) and preschoolers (aged 3–6 years) at home; the findings are shown in Table 8. The percentages of children exhibiting hand-to-mouth and object-to-mouth activities were 54.29% and 45.71% in the toddler group and 61.61% and 23.21% in the preschooler group, respectively. The younger group reported a higher frequency and longer duration of mouthing behaviors than the preschooler group.

### 3.6. Soil and Dust Ingestion

The concentrations of tracers in feces, household dust, and neighborhood soil of 113 preschoolers aged ≤6 years were analyzed. The daily dry weight of feces was 8.63 g/day for all children with values of 8.85 g/day for boys and 8.42 g/day for girls. The concentrations of aluminum, silica, titanium, and zirconium in the children’s feces, neighborhood soil, and dust were analyzed, as shown in Table 9. We also estimated soil and dust ingestion separately without and/or with outdoor time–activity patterns using four elements (Table 10). Aluminum was selected because the maximum amount of soil ingested corresponded with the lowest estimate among the tracers based on the limiting tracer method. This resulted in a 29.3% difference between the total soil and dust ingestion rate (70.63 mg/day) and dust only ingestion rate (20.68 mg/kg, three children aged <12 months). The predicted amount of total soil and dust ingestion was approximately 70 mg of dust, while the amount of ingestions via soil and dust deposits were both 35 mg per day.

## 4. Discussion

The risk assessment of hazards in children’s products should be conducted based on exposure scenarios that incorporate the ages, body weights, BSAs, and behavioral patterns of children who use the products [20]. In general, children’s body weight is proportional to the inhalation rates and BSAs, which may result in differences in the body burden of chemicals even when identical amounts are absorbed in their bodies.

Although the difference in age grouping among countries restricts the direct comparison of body weights across all ages, we identified that the average body weight of Korean children aged 16–18 years is 62.4 kg (67.3 kg for boys and 56.8 kg for girls). These values are lower than the average body weights of Australian children of the same age group, which were 75.0 kg for boys and 61.9 kg for girls [21]. As average exposures to chemicals are standardized based on the average body weight of the relevant age group, applying the body weight values of Australia and other Western countries to Korean children may result in the underestimated exposure levels [22].

Inhalation rates are required to assess the level of exposure to hazardous substances absorbed in the respiratory system. Because it is not possible to directly measure the inhalation rates in participants who are wearing gas analyzers on a daily basis, the inhalation rate has been generally measured through heart rate monitoring, metabolic equivalent task, and doubly labeled water method [23]. The average inhalation rate of Korean children aged 5–6 years was estimated to be 10.8 m^3^/day, which is similar to that of European children aged 4–6 years (11 m^3^/day) [24]. The average inhalation rate of Korean children was relatively similar to that of Japanese children (9.9 m^3^/day) [25]. On the contrary, the average inhalation rate of Korean children was lower than that of US children (12.16 m^3^/day) [25]. These differences among countries may be attributable to the difference in gas analyzers used and in the measurement methods for metabolic equivalent task and inhalation rate. However, the inhalation rates in Korean and Japanese children are relatively lower than those in children from Western countries.

With regard to food intake, we divided the foods into 14 groups (e.g., grains, seaweed, vegetables, nuts) to determine the dietary characteristics of Korean children. Meat consumption per body weight among Korean children ranged from 3.06 g/kg/day to 4.16 g/kg/day, which increased to 38% to 87% in the past 15 years. The daily intakes of grains, vegetables, and fruits among US children aged 1–2 years were 6.4 g/kg/day, 6.7 g/kg/day, and 7.8 g/kg/day, respectively [26]. The average intake of fish/shellfish in Korean preschoolers was 2.26–3.00 g/kg/day, which was approximately 9.4–11.5 times higher than that of the corresponding American children [26]. On the contrary, Korean children consumed only 47.6%–53.6% of the total percentage of dairy products consumed by US children. The consumption rates of vegetables and fish/shellfish were similar between Korean and Japanese children ranging from 144.1 to 238.6 g/day and from 40.3 to 75.2 g/day, respectively; however, the fruit intake of Korean children was two times higher than that of Japanese children ranging from 91.9 to 104.3 g/day [27].

The time–activity patterns of the population constitute the most essential data for assessing the level of exposure to environmental pollutants. The duration and frequency of exposure depend on the activity patterns of individuals and the time spent at the location of each activity [28]. In one of the exposure assessment methods, the concentration of air pollutants in microenvironments and the time spent therein were applied to a mathematical model [29]. On weekdays, Korean children aged ≤9 years and 10–18 years spent 16.86 h and 12.92 h in houses and 0.72 h and 2.30 h in other indoor locations, including private tutoring institutions, respectively. The total time spent indoors by German, American, and Canadian adolescents was 20.6 h, 20.93 h, and 20.98 h, respectively [30,31]. The Korean children spent more time indoors (22.62 h) than children from other countries, which suggests the importance of managing indoor air quality at main indoor locations such as homes, kindergartens, and schools. The time spent outdoors by Korean children aged 3–9 years was 0.73 h, which was a weighted average of weekdays and weekend. The time spent outdoors was only 36.7%–43.7% of the average times spent outdoors by US children aged 3–10 years and Canadian children aged ≤10 years, which are 1.99 and 1.67 h, respectively [31].

The average hand-to-mouth and object-to-mouth contacts per hour for Korean children between 1 and 2 years were 2.17 and 2.91, respectively. The average duration of object-to-mouth activities was 8.5 min/h, which was similar to the value of 8.9 min/h (7.3–10.5 min/h) reported by Greene’s study [32]. The object-to-mouth contact frequency was one-third lower in the 3-6 year age group than in the corresponding US (10 contacts/h) and Taiwanese (13.8 contacts/h) children groups [33]. To minimize the possible uncertainties caused by a recall-based survey, we also recorded children’s object-to-mouth and hand-to-mouth activities; the recorded samples were greater than those reported in similar studies conducted in other countries. However, video recording was conducted only once per day for 30 min, which may have limitations in terms of replicability. Despite these limitations, the reasons for the relatively low values might be as follows: children spent a significant amount of time playing computer games indoors [34] and Korean mothers specifically prohibit their children from putting their hands into their mouth.

One of the main exposure routes of pollutants in young children is non-dietary exposure such as soil and dust ingestion. The US EPA reported that the unintentional soil and dust intake by children aged between 1 and 6 years is 80 mg/day, which consisted of soil intake of 40 mg/day and dust intake of 40 mg/day [35]. Using EPA’s stochastic exposure and dose simulation model for multimedia pollutants, the mean and 95th percentile of the amount of ingested of soil and dust were predicted through children mouthing patterns and were found to be 68 and 224 mg/day for children aged 3 to <6 years, respectively [36]. These values were similar to the corresponding values obtained from this study (70 mg/day as mean, 200 mg/day as the 95th percentile for Korean children aged <7 years). The Korean Exposure Factors Handbook by the MOE reported that children unintentionally ingest 118 mg/day dust and soil, while this study estimated a non-dietary ingestion amount of 70 mg/day, indicating a gradual decrease in soil and dust ingestion when compared with that via other sources [8]. These findings may be partially attributable to the increasing percentage of playgrounds with urethane floors.

This study has some limitations. First, we only estimated the amount of soil and dust ingestion based on the concentration of tracers in children’s feces; hence, the contribution of tracers through foods and their bioavailability in human bodies were not considered. Second, a long-term investigation was not conducted on a large number of participants due to inherent characteristics and various physiological uncertainties. However, these limitations are common among previous international studies.

Despite some limitations, exposure factors for Korean children could be used as the most fundamental and critical data for estimating the level of exposure to environmental pollutants. Our findings were based on the body weights/BSAs of 6700 children, the food intake of 6000 children, and the time–activity patterns of 9100 children, which represent the reliability among the recently available data in terms of the representativeness of population groups, sample sizes, and being up-to-date. Additionally, while previous studies estimated the breathing rates from heart rates and metabolic equivalents, we significantly reduced the measurement errors by measuring inhalation rates using gas analyzers and treadmills. However, the recommended values for mouthing activity and soil and dust ingestion, which are the main exposure routes in children, were limited considering the lack of accessibility and high cost of requirements.

The US EPA’s Exposure Factors handbook has consistently updated its contents with the latest data to revise recommendations and to add susceptible populations over recent decades. In the future, these factors need to be continuously updated, along with more extensive survey areas and improved measurement methods for inhalation rate, soil and dust ingestion, and other factors.

## 5. Conclusions

With the aim to develop general exposure factors, a critical requirement for exposure and risk assessment in Korean children aged ≤18 years, we utilized the existing data from the KNHNES and the Statistics Korea for body weight, BSAs, and food and water intake. Other exposure factors such as inhalation rate, time–activity pattern, soil and dust ingestion, and hand-to-mouth behaviors were assessed by performing measurements and surveys. These general exposure factors are expected to be used in various studies in the field of exposure and environmental health risk assessment. The Korean Exposure Factors Handbook for children contains a detailed description of methods and recommended values across five categories, which can be downloaded at the Environmental Information Library of the National Institute of Environmental Research (http://library.nier.go.kr).

## Figures and Tables

**Figure 1 ijerph-17-01988-f001:**
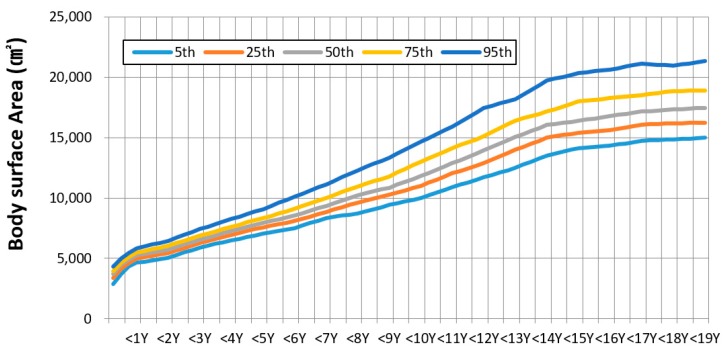
Estimation of body surface area based on body weight and height according to age.

**Figure 2 ijerph-17-01988-f002:**
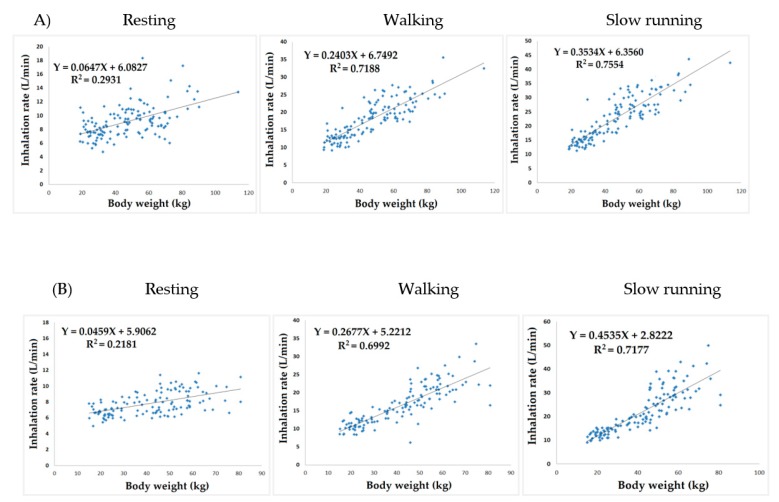
Estimated short-term inhalation rates for boys (**A**) and girls (**B**) aged below 4 years using the correlation between inhalation rate and body weight of children aged 5–18 years.

**Table 1 ijerph-17-01988-t001:** Recommended values and statistics for body weight according to age group.

Sex	Age Group	N	Body Weight (kg)						
			Mean *	S.D.	5^th^	25th	50th	75th	95th
**Boys**	Birth to <3 months	51	5.9	1.1	4.1	5.1	6.1	6.7	7.6
	3 to <6 months	50	8.0	0.9	6.3	7.5	8.0	8.5	9.3
	6 to <9 months	51	8.9	1.2	7.2	8.0	8.8	9.6	11.5
	9 to <12 months	50	10.1	1.0	8.4	9.5	10.0	10.9	12.0
	1 to <3 years	344	12.6	5.2	9.6	11.0	12.2	13.5	15.4
	3 to <7 years	798	19.2	4.3	13.7	16.2	18.5	21.5	27.4
	7 to <10 years	560	30.9	7.4	21.6	25.9	29.0	34.6	44.9
	10 to <13 years	579	44.7	10.9	30.1	36.2	43.2	51.0	65.7
	13 to <16 years	547	61.5	13.7	42.7	51.1	58.9	68.8	85.3
	16 to <18 years	453	67.3	14.0	50.3	58.3	64.7	74.7	93.1
Girls	Birth to <3 months	50	5.2	0.9	3.7	4.6	5.2	5.8	6.4
	3 to <6 months	51	7.0	1.0	5.5	6.3	7.1	7.7	8.2
	6 to <9 months	50	8.4	0.9	7.2	7.4	8.3	9.0	9.7
	9 to <12 months	53	9.2	1.0	7.7	8.4	9.2	9.9	11.0
	1 to <3 years	356	11.8	1.8	8.9	10.4	11.7	12.9	14.9
	3 to <7 years	717	18.4	3.8	13.5	15.6	17.8	20.4	25.4
	7 to <10 years	554	29.0	7.0	20.6	24.1	27.7	32.3	43.5
	10 to <13 years	469	42.7	10.0	28.4	35.9	41.5	47.7	60.5
	13 to <16 years	473	52.8	9.7	39.7	46.2	51.2	57.5	69.0
	16 to <18 years	440	56.8	9.8	44.3	50.1	54.8	62.0	76.3
Both sex combined	Birth to <3 months	101	5.6	1.03	3.7	4.9	5.6	6.2	7.3
	3 to <6 months	101	7.5	1.02	5.7	6.8	7.6	8.1	8.9
	6 to <9 months	101	8.7	1.07	7.1	7.9	8.6	9.3	10.2
	9 to <12 months	103	9.7	1.11	7.9	8.9	9.6	10.3	11.4
	1 to <3 years	700	12.2	3.92	9.2	10.7	11.9	13.2	15.1
	3 to <7 years	1515	18.8	4.05	13.6	16.0	18.1	21.0	26.6
	7 to <10 years	1114	30.0	7.29	21.0	24.9	28.3	33.2	44.3
	10 to <13 years	1048	43.8	10.52	29.3	36.0	42.7	49.6	63.9
	13 to <16 years	1020	57.4	12.72	40.5	48.5	54.6	63.5	81.0
	16 to <18 years	893	62.4	13.27	45.7	53.0	59.8	69.4	88.2

* Arithmetic mean; S.D.: standard deviation.

**Table 2 ijerph-17-01988-t002:** Recommended values and statistics for body surface area according to age group.

Sex	Age Group	Body Surface Area (cm^2^)						
		Mean *	S.D.	5th	25th	50th	75th	95th
**Boys**	Birth to <3 months	3806	457	3042	3464	3901	4120	4450
	3 to <6 months	4618	312	3982	4430	4621	4808	5111
	6 to <9 months	4985	423	4372	4674	4952	5217	5830
	9 to <12 months	5400	360	4839	5177	5371	5648	6058
	1 to <3 years	6241	593	5307	5800	6258	6692	7209
	3 to <7 years	8426	1120	6843	7626	8288	9146	10,514
	7 to <10 years	11,402	1480	9347	10,378	11,112	12,204	14,140
	10 to <13 years	14,303	1915	11.492	12,788	14,082	15,507	17,906
	13 to <16 years	17,464	2048	14.317	15,979	17,192	18,679	20,777
	16 to <18 years	18,436	1958	15,930	17,182	18,158	19,644	21,878
Girls	Birth to <3 months	3538	368	2867	3284	3551	3787	4062
	3 to <6 months	4262	355	3671	3998	4290	4522	4730
	6 to <9 months	4799	348	4343	4455	4775	5016	5263
	9 to <12 months	5126	364	4561	4857	5131	5371	5704
	1 to <3 years	6084	584	5101	5640	6089	6501	7043
	3 to <7 years	8234	1026	6813	7447	8127	8851	10,139
	7 to <10 years	11,002	1449	9058	9957	10,819	11,808	13,897
	10 to <13 years	13,973	1803	11,189	12,650	13,906	15,045	17,120
	13 to <16 years	15,908	1517	13,733	14,862	15,725	16,753	18,421
	16 to <18 years	16,561	1489	14,543	15,507	16,343	17,397	19,469
Both sex combined	Birth to <3 months	3691	430	2865	3391	3700	3944	4348
	3 to <6 months	4442	376	3743	4169	4471	4675	4985
	6 to <9 months	4905	348	4317	4629	4882	5115	5417
	9 to <12 months	5278	359	4640	5004	5255	5483	5837
	1 to <3 years	6162	594	5201	5732	6163	6583	7143
	3 to <7 years	8336	1081	6829	7518	8223	9038	10,365
	7 to <10 years	11,204	1479	9188	10,158	10,923	11,969	13,981
	10 to <13 years	14,146	1870	11,367	12,736	14,040	15,293	17,529
	13 to <16 years	16,727	1976	13,897	15,348	16,405	17,812	20,297
	16 to <18 years	17,558	1988	14,865	16,139	17,283	18,752	21,169

* Arithmetic mean; S.D.: standard deviation.

**Table 3 ijerph-17-01988-t003:** Short-term inhalation rate (m^3^/h) for boys and girls stratified by activity levels.

Age	Boys			Girls			
	Resting ^(^^a)^	Walking ^(^^b)^	Slow Running ^(^^c)^	Resting ^(^^d)^	Walking ^(^^e)^	Slow Running ^(^^f)^	*p*-Value
5	0.50 ± 0.15	0.77 ± 0.17	0.86 ± 0.16	0.40 ± 0.06	0.58 ± 0.08	0.65 ± 0.10	Significantly difference between (a) and (d), (b) and (e), and (c) and (f) at *p*-value of <0.05
6	0.46 ± 0.06	0.73 ± 0.10	0.81 ± 0.06	0.42 ± 0.06	0.69 ± 0.10	0.77 ± 0.06	
7	0.46 ± 0.05	0.75 ± 0.10	0.89 ± 0.14	0.36 ± 0.12	0.61 ± 0.22	0.72 ± 0.26	
8	0.47 ± 0.09	0.80 ± 0.14	0.96 ± 0.16	0.42 ± 0.05	0.74 ± 0.15	0.86 ± 0.16	
9	0.49 ± 0.11	0.80 ± 0.09	0.97 ± 0.10	0.44 ± 0.07	0.72 ± 0.11	0.87 ± 0.14	
10	0.47 ± 0.10	0.82 ± 0.17	1.00 ± 0.20	0.40 ± 0.16	0.73 ± 0.33	0.91 ± 0.37	
11	0.48 ± 0.08	0.94 ± 0.22	1.18 ± 0.28	0.54 ± 0.19	0.87 ± 0.18	1.03 ± 0.17	
12	0.57 ± 0.09	1.07 ± 0.17	1.33 ± 0.20	0.50 ± 0.09	0.96 ± 0.16	1.17 ± 0.16	
13	0.61 ± 0.05	1.24 ± 0.14	1.56 ± 0.15	0.49 ± 0.12	1.05 ± 0.20	1.39 ± 0.26	
14	0.66 ± 0.13	1.35 ± 0.19	1.56 ± 0.61	0.48 ± 0.10	1.19 ± 0.23	1.72 ± 0.31	
15	0.61 ± 0.10	1.34 ± 0.28	1.78 ± 0.39	0.52 ± 0.10	1.23 ± 0.15	1.68 ± 0.23	
16	0.65 ± 0.22	1.41 ± 0.31	1.94 ± 0.34	0.50 ± 0.08	1.43 ± 0.26	2.10 ± 0.43	
17	0.57 ± 0.21	1.43 ± 0.17	1.94 ± 0.19	0.52 ± 0.08	1.36 ± 0.18	1.99 ± 0.30	
18	0.594 ± 0.116	1.381 ± 0.237	1.855 ± 0.253	0.492 ± 0.074	1.393 ± 0.209	2.082 ± 0.281	

* Resting (a) and (d) defined as sitting; walking (b) and (e) at speed level 2–3 km/h; slow running (c) and (f) at speed level 3–4 km/h.

**Table 4 ijerph-17-01988-t004:** Recommended values for inhalation rate according to age group.

Sex		Birth to 3 Years	3 to <7 Years	7 to <10 Years	10 to <13 Years	13 to <16 Years	16 to <18 Years	Total
Boys	m^3^/day (m^3^/kg/day)	9.75 (0.87)	10.96 (0.57)	11.65 (0.40)	13.24 (0.30)	15.98 (0.27)	16.15 (0.24)	13.64 (0.39)
Girls	m^3^/day (m^3^/kg/day)	9.23 (0.90)	9.78 (0.53)	10.20 (0.36)	12.19 (0.29)	12.71 (0.24)	13.73 (0.24)	11.78 (0.38)
Both sex combined	m^3^/day (m^3^/kg/day)	9.49 (0.89)	10.38 (0.55)	10.93 (0.38)	12.74 (0.30)	14.38 (0.26)	14.98 (0.24)	12.73 (0.39)

**Table 5 ijerph-17-01988-t005:** Average daily intake (g/day) and consumption rate (%) of major food.

Food Groups		1 to <3 Years (n = 670)	3 to <7 Years (n = 1436)	7 to <13 Years (n = 2036)	13 to <18 Years (n = 1681)	Total (n = 5823)
Grain products	Mean	279.83	388.52	533.13	585.18	498.71
	(Rate, %)	(99.4)	(99.9)	(99.9)	(98.3)	(99.7)
Vegetables	Mean	107.89	156.80	235.55	291.09	228.11
	(Rate, %)	(98.8)	(99.6)	(99.4)	(98.9)	(99.2)
Fruit products	Mean	136.06	173.91	174.96	134.63	155.74
	(Rate, %)	(81.6)	(81.5)	(78.2)	(64.3)	(75.4)
Meat products	Mean	48.41	71.25	137.09	177.62	130.33
	(Rate, %)	(86.0)	(91.4)	(92.9)	(90.0)	(90.9)
Egg products	Mean	21.31	30.12	36.33	31.97	31.95
	(Rate, %)	(66.7)	(73.9)	(74.8)	(66.4)	(71.2)
Fish/shellfish	Mean	35.27	41.83	58.21	58.97	52.90
	(Rate, %)	(79.1)	(87.0)	(83.5)	(74.8)	(81.3)
Seaweeds	Mean	12.64	18.91	18.97	13.86	16.42
	(Rate, %)	(60.0)	(64.7)	(56.7)	(43.0)	(55.1)
Dairy products	Mean	242.55	234.11	217.39	158.72	200.97
	(Rate, %)	(90.2)	(84.9)	(79.8)	(64.2)	(77.7)
Seasonings	Mean	9.94	21.77	40.62	55.33	39.37
	(Rate, %)	(94.8)	(99.4)	(99.1)	(97.9)	(98.3)
Sugar products	Mean	5.69	8.77	13.88	13.42	11.86
	(Rate, %)	(70.8)	(87.3)	(88.5)	(81.4)	(84.1)
Fabricated foods	Mean	2.62	7.70	15.91	23.87	15.96
	(Rate, %)	(6.0)	(10.2)	(14.7)	(16.3)	(13.1)
Drinking water *	Mean	440	599	825	952	790
	(Rate, %)	(100.0)	(99.9)	(100.0)	(99.7)	(99.9)

* mL/day.

**Table 6 ijerph-17-01988-t006:** Average and 95th percentile values of consumer-only dietary intake (g/day) by age group.

Food Groups		1 to <3 Years	3 to <7 Years	7 to <13 Years	13 to <18 Years	Total
Grain products (cooked)	Mean	281.99	388.78	534.54	589.54	500.97
	(95th)	(579)	(711)	(967)	(1179)	(1012)
Vegetables	Mean	109.04	157.55	237.17	294.78	230.16
	(95th)	(337)	(410)	(560)	(749	(580)
Fruit products	Mean	167.49	214.27	222.96	210.27	211.01
	(95th)	(452)	(635)	(712)	(756)	(680)
Meat products	Mean	56.95	78.04	147.62	199.93	144.39
	(95th)	(219)	(240)	(476)	(651)	(476)
Egg products	Mean	32.16	40.53	48.74	49.11	45.57
	(95th)	(95)	(117)	(143)	(141)	(132)
Fish/shellfish	Mean	45.08	48.39	70.28	80.15	66.43
	(95th)	(174)	(187)	(275)	(303)	(262)
Seaweeds	Mean	21.11	29.5	33.75	33.26	31.16
	(95th)	(63)	(85)	(105)	(99)	(99)
Dairy products	Mean	270.39	276.44	276.8	253.57	268.6
	(95th)	(696)	(624)	(637)	(633)	(637)
Seasonings	Mean	10.46	21.9	41.03	56.61	40.08
	(95th)	(30)	(62)	(106)	(167)	(126)
Sugar products	Mean	7.96	10.11	15.77	16.6	14.21
	(95th)	(33)	(45)	(65)	(66)	(58)
Fabricated foods	Mean	42.87	68.69	104.18	149.09	115.36
	(95th)	(107)	(237)	(288)	(378)	(311)
Drinking water *	Mean	439.95	599.52	825.71	954.67	790.83
	(95th)	(1000)	(1200)	(2000)	(2000)	(2000)

* mL/day.

**Table 7 ijerph-17-01988-t007:** Recommended values (hours) for activity factors by weekday and weekend.

	Age Group	Indoor			Total Indoors	Outdoors	In-Transit
		Home	Childcare Center/School *	Others			
Weekdays	Birth to <3 years (n = 704)	18.40	3.98	0.52	22.90	0.45	0.65
	3 to <7 years (n = 855)	16.15	5.90	0.62	22.67	0.55	0.78
	7 to <10 years (n = 521)	15.93	5.42	1.18	22.53	0.59	0.89
	10 to <13 years (n = 1,434)	14.04	6.24	2.39	22.67	0.85	0.48
	13 to <16 years (n = 1,476)	12.98	7.29	2.38	22.65	0.75	0.60
	16 to <18 years (n = 1,232)	11.82	8.47	2.14	22.43	0.61	0.96
Weekends	Birth to <3 years (n = 704)	20.56	0.13	1.44	22.13	0.98	0.89
	3 to <7 years (n = 855)	19.75	0.23	1.84	21.82	1.16	1.02
	7 to <10 years (n = 521)	19.77	0.22	1.89	21.88	1.12	1.00
	10 to <13 years (n = 962)	17.85	0.26	4.51	22.62	0.39	0.99
	13 to <16 years (n = 1,002)	18.33	0.31	4.20	22.84	0.50	0.66
	16 to <18 years (n = 786)	16.85	1.30	4.51	22.66	0.54	0.80

* The average hours per day at childcare center (<7 years old) or school (≥7 years old).

**Table 8 ijerph-17-01988-t008:** Mouthing activity of Korean children aged 1–6 years.

Parameters	Age Group (Years)	Respondents (%)	Frequency (Contacts/Hour)		Duration (Seconds/Contact)	
			Mean	95th	Mean	95^th^
Hand-to-mouth	1 to <3 years (n = 35)	54.29	2.17	10.40	1.22	5.12
	3 to <6 years (n = 112)	61.61	3.09	12.05	2.34	5.84
	Total (n = 147)	59.86	2.87	5.00	2.07	60.00
Object-to-mouth	1 to <3 years (n = 35)	45.71	2.91	22.20	2.87	23.00
	3 to <6 years (n = 112)	23.21	1.03	5.35	0.65	4.00
	Total (n = 147)	28.57	1.48	5.00	1.18	30.00

**Table 9 ijerph-17-01988-t009:** Concentration (mg/kg, n = 113) of tracers in children’s feces, neighborhood soil, and indoor dust at home.

Media	Tracer *	Mean	S.D.	Median	Min.	Max.
Faces	Al	117.3	271.1	62.9	10.27	2695
	Si	334.3	523.1	259.7	33.83	5476.33
	Ti	30.4	153.8	4.8	1.63	1520.17
	Zr	4.2	11.0	1.6	0.23	98
Neighborhood soil in outdoor	Al	13,864.33	6971.7	13216	3522.4	35588
	Si	883.21	671.03	627.2	86.8	2369.92
	Ti	626.76	445.77	522.62	33.6	3130.40
	Zr	11.24	8.95	8.68	1.96	58.24
Settle house dust in indoor	Al	12,800.50	11,396.11	8818.18	2689.48	64,279.52
	Si	14,380.62	17,903.26	7230.99	1726.47	98,357.14
	Ti	1118.86	1483.05	662.80	133.14	11,130.83
	Zr	472.83	1438.71	89.01	4.44	14,238.75

* Al, Aluminum; Si, Silica; Ti, Titanium; Zr, Zirconium.

**Table 10 ijerph-17-01988-t010:** Estimation of soil and dust ingestion (mg/day) based on outdoor activities.

Tracer	All Participants (n = 113)		Participants Engaging in Outdoor Activities (n =110)	
	Mean	95th	Mean	95th
Aluminum	68.46	214.85	70.63	217.56
Silica	5809.19	22,073.99	6047.57	22,589.11
Titanium	655.17	1507.85	682.63	1512.01
Zirconium	2878.65	10,466.67	2961.53	10,925.00

* Al, Aluminum; Si, Silica; Ti, Titanium; Zr, Zirconium.

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
