# Peer review of "Development of General Exposure Factors for Risk Assessment in Korean Children"

_ijerph, 2020, doi:10.3390/ijerph17061988_

Round 1
Reviewer 1 Report
Review of manuscript titled “Development of General Exposure Factors for Risk Assessment in Korean Children” IJERPH_704724
Overall: Authors conducted studies to collect exposure factors for Korean children. As some exposure factors (e.g., time-activity patterns, food consumption) may differ by countries and regions, this study may add new information to the field of study and can serve as a foundation for exposure and risk assessment of Korean children. Overall, the manuscript is well-written, clear, and well-organized. Before being accepted for publication, there are some minor changes to be made to deliver clear content to readers.
Introduction
L61: Suggest changing the text: “This study… for exposure factors by..”.
Methods
L92: Not clear whether NHANES is US NHANES or Korea NHANES. Please specify because authors also mentioned Korea NHANES in line 119. Suggest using abbreviation of NHANES.
L96: Not sure whether all first letters should be capital for BSA.
L134: “nationwide rood proportional allocation” I suggest revising this or provide more information about this method.
L161: Suggest deleting the text of “using the estimation equation”.
L177: Suggest revising the text because humans or children are not exposed to factors.
L178: It is not clear. The suggested text is the following: “Ultimately, the recommended values for soil and dust ingestion were calculated by accounting for the background concentration of each tracer”. Is this what authors meant?
Results
L187-190: This is method, so is redundant to be included.
L196: Same as above.
L198: Figure 1 does not show an exponential curve. Do you mean quickly increasing?
L199: Figure 1 does not show decrease. Do you mean slowly increasing?
L205: Figure 1 does not have a legend to indicate the body weight that corresponds to each line.
L210: Authors mentioned a t-test in the statistical analysis section, but I cannot see the results of the t-test in Figure 2. So, authors cannot make a statement such as “significantly”.
L217: It would be much better if sub-figures have a title of walking and slow running instead of Step 1 and Step 2.
L221: Please increase the font size for all axis titles and regression equations, R2 values. Please use a title of walking and slow running instead of Step 1 and Step 2.
L225: What is “minutely breathing rates”?
L228: Table 3: Reduce or remove space between “Box sex” and “combined” in the first column.
L237: Suggested text: “The average intake of fabricated foods for consumers only (115.36 g/day) was seven times larger than that for all children (15.96 g/day).”
Table 6: What is * in the fourth column? Please specify in the footnote of the table.
Table 8 title: A comma is missing after feces. “The concentration (mg/kg, n = 113) of tracers in children’s feces, neighborhood soil, and indoor residential dust”
Table 9: You cannot calculate mean and 95th percentile with 3 samples!! Please remove this or reconsider merging to the next column.
Discussion
L289: Suggested text: “In general, children’s body weight is proportional to inhalation rates and BSAs, which may result in different body burden of chemicals...”
L291-294: It is a long sentence. Please break into two sentences.
L296: Suggested text: “…, applying the body weight values of Australia and other Western countries to Korean children may result in underestimated exposure.”
L308: This sentence is awkward. Suggested text: “…, the inhalation rates of Korean and Japanese children are relatively lower.” Or “…, the inhalation rates of Korean and Japanese children might be lower.”
L310: “(grains, seaweed, vegetables, nuts)” can be removed or add “e.g.,” in front of grains.
L311-313: This sentence is awkward. Do you mean “The meat consumption per body weight ranged from 3.06 g/kg/day to 4.16 g/kg/day, which were increased 30% to 150% over the past 15 years in Korean children.”?
L314: Do you mean “The average intake of fish and shellfish…”?
L318: Revise this text: “The vegetables, fish and shellfish…”
L318: I don’t think that the values are similar each other: 144.1 to 238.6 g/day and 40.3 to 75.2 g/day. Please revise this.
L331: Adding additional information would make this sentence clearer. “The time spent outdoors of Korean children aged x-x years was…”
L338: It is “Taiwanese”.
L345: Each person has one mouth. So, remove ‘s’ from “their mouths”.
L346: Suggested text: “One of the important exposure routes for young children is non-dietary dust and soil ingestion”.
L363: I am not clear how suitability was tested and verified in this study.
L366: “the highest reliability” I think authors did not compare various reliabilities and reached this conclusion.
Conclusions
L378, 383: Please use “exposure and risk assessment” instead of “risk assessment” because risk assessment is the prior step of exposure assessment.
L379: Please specify whether this is U.S. NHANES or Korea NHANES.
L384: Because authors used uppercase letters for Korean Exposure Factors Handbook in line 353, please revise here too.
Minor: In the funding statement, a word “Funding” is redundant.
Reviewer 2 Report
The substantial part of the manuscript could be translation from Korean article,
http://www.koreascience.or.kr/article/JAKO201719951668263.page
Journal of Environmental Health Sciences
Volume 43 Issue 3 / Pages.167-175
by the authors.
I cannot read Korean language, but can find similar or identical numbers and figures.
Authors may include new data, but did not at all mention about previous study and differences from it.
Nowadays, submission of simple translation can be deemed as duplicate publication. It poses problems
regarding with research ethics and, also, copyrights. At least,
authors should confirm and obtain the permission for secondary publication, from parties concerned. Before the issue will be closed, I cannot recommend it for publication.
I recommend authors to compare the results from previous publication.
Also, on identical figure and table, copyright permission should be obtained from Journal of Environmental Science (Korea).
Then, I recommend authors to revise texts and captions according to the comparison and to clarify the paper is revision of previous paper.
Additional comment is on significant figures of data.
Authors showed 4 to 6 digits but it is not appropriate for
usual measurement precision.
Round 2
Reviewer 2 Report
I am satisfied with the revision and have an minor opinion.
The sentence; “We have been studying the general exposure factors in Korean children since 2013, and the results were published in the 2017 Korean domestic edition [19]. In contrast, this study was carried out for international comparative evaluation by adding updated national statistics and new experiments, such as those for the rates of soil and dust ingestion and unintentional exposure through hand-to-mouth contact”, is better to be moved to section of 'Introduction'.
